# Knowledge, Attitude, and Acceptance of Sinopharm and AstraZeneca’s COVID-19 Vaccines among Egyptian Population: A Cross-Sectional Study

**DOI:** 10.3390/ijerph192416803

**Published:** 2022-12-14

**Authors:** Marian S. Boshra, Marwa O. Elgendy, Lamiaa N Abdelaty, Mahmoud Tammam, Abdullah S. Alanazi, Abdulaziz Ibrahim Alzarea, Saud Alsahali, Rania M. Sarhan

**Affiliations:** 1Clinical Pharmacy Department, Faculty of Pharmacy, Beni-Suef University, Beni-Suef 62511, Egypt; 2Clinical Pharmacy Department, Beni-Suef University Hospitals, Faculty of Medicine, Beni-Suef University, Beni-Suef 62511, Egypt; 3Department of Clinical Pharmacy, Faculty of Pharmacy, Nahda University (NUB), Beni-Suef 62513, Egypt; 4Department of Clinical Pharmacy, Faculty of Pharmacy, October 6 University, Giza 12525, Egypt; 5IQVIA, Cairo 10245, Egypt; 6Department of Clinical Pharmacy, College of Pharmacy, Jouf University, Sakaka 72388, Saudi Arabia; 7Health Sciences Research Unit, Jouf University, Sakaka 72388, Saudi Arabia; 8Clinical Pharmacy Department, College of Pharmacy, Jouf University, Sakaka 72388, Saudi Arabia; 9Department of Pharmacy Practice, Unaizah College of Pharmacy, Qassim University, Unaizah 51911, Saudi Arabia

**Keywords:** COVID-19, vaccines, awareness, Sinopharm, AstraZeneca

## Abstract

Background: This study aimed to evaluate the Egyptian population’s preference and awareness related to available COVID-19 vaccines and to determine different factors that can affect beliefs concerning these vaccines. Methods: A cross-sectional web-based study was carried out among the general population in Egypt. Data collection was conducted via an online questionnaire. Results: About 426 subjects participated in the survey. Vaccine preference is nearly equally even (50%) among all respondents. There was no significant difference in vaccine preference according to age, gender, residence, educational level, or social status. About 50% of public respondents mentioned that both AstraZeneca and Sinopharm vaccines do not offer protection against new variant COVID-19 strains. Healthcare workers are the lowest respondents to agree that vaccines offer protection against new COVID-19 variants (10.9%) compared to unemployed respondents (20.3%) and other professions (68.8%) with a statistically significant difference (*p* < 0.005). Safety of vaccine administration among children below 18 showed statistical differences for gender and educational level predictors. Conclusions: Most of the study population has satisfying knowledge about the COVID-19 vaccine. Continuous awareness campaigns must be carried out so that the people’s background is updated with any new information that would help in raising the trust in vaccination.

## 1. Introduction

The outbreak of the coronavirus disease 2019 (COVID-19) is responsible for a number of threats and dangers to human health, including increased death and morbidity rates around the globe [1,2]. Therefore, the development of effective vaccines against COVID-19 became the main target in many clinical trials. Vaccine distribution requires effective, sufficient, and accurate health system strategies to achieve high rates of vaccine acceptance and trust to those who deliver it [3]. In many clinical trials, numerous types of vaccinations were reported to be more than 90% effective against COVID-19 infection [4].

Rapid development and challenges in vaccine production have become a continuous, global and critical necessity following the medical decision-based agreement that the benefits greatly outweigh the risks [5]. In 2019, the risk of COVID-19 pandemic and vaccine indecision were both hazardous to global health that was declared by the world health organization (WHO) [6]. Currently, the whole world is forcing both hazards [7].

Egypt began receiving anti-COVID-19 vaccinations such as Sinopharm (BBIBP-CorV), AstraZeneca vaccine, and Sputnik V in December 2020 [8]. As of March 2021, Egypt has begun distributing COVID-19 vaccines. Although the country wants to vaccinate 40 percent of its people against COVID-19 by the end of 2021, Egypt is one of the countries suffering from vaccine hesitancy [9]. According to the WHO, by 24 June 2021, a total of 2,624,733,776 doses of COVID-19 vaccinations had been administered globally, and by 23 June 2021, a total of 4,138,935 doses had been administered in Egypt (World Health Organization) (WHO) [10].

Awareness of the causes of acceptance or rejection of vaccine and determination of the reasons why people may hesitate to take the vaccine against COVID-19 are important issues to address. This can be a helpful tool to different healthcare systems all over the world to increase the rates of vaccine acceptance to control the spread of the disease and to achieve the possible safety for all healthcare professionals [11,12]. The reported efficacy of different types of COVID-19 vaccines had an impact on its acceptance, especially for those that were reported to have relatively low efficacy [13].

To obtain herd immunity, it was estimated that at least 55% of the population must accept the COVID-19 vaccination, with estimates reaching as high as 85% depending on the infection rate and country [14,15]. The acceptance of vaccines is a complex decision-making process based on many factors, for example, awareness about the difference in efficacy of vaccines, the mode of action, the degree of protection and immunity that can be achieved with receiving the vaccine, and the risk of side effects that may be demonstrated with different doses of vaccines

With the presence of vaccine hesitancy in the public, it is difficult to reach the required vaccination levels [15]. This hesitancy may be attributed to misinformation about the spread of COVID-19 across media. Therefore, the Egyptian public health officials and politicians have been starting awareness plans and health communication programs for presenting effective messaging about the importance and the safety of the vaccine to prevent future infections and deaths [16]. Consequently, higher rates of acceptance regarding COVID-19 vaccines among the Egyptian population are required to attain a successful immunization program. To achieve this, it is important to determine certain issues, including estimating Egyptian risk perceptions and fears about the vaccine, the acceptance rate of vaccine decisions, and increasing the trusted media communication sources [17,18]. In addition, the Egyptian government makes significant efforts to ensure that sufficient quantities of COVID-19 vaccines are available for the Egyptian people [8]. Moreover, it plays a significant role by prioritizing the vaccination of healthcare workers and older persons with chronic conditions [16].

The aim of this study is to establish the general population’s awareness and preference for COVID-19 vaccines in Egypt, as well as to identify the determinants of their attitudes that can be leveraged to boost vaccine uptake and fast achieve the necessary herd immunity.

## 2. Materials and Methods

### 2.1. Study Design

A cross-sectional and observational web-based survey was conducted to determine the knowledge and awareness about the COVID-19 vaccine and the potential of different psychological factors of COVID-19 vaccine hesitancy among the general population in Egypt. The survey was conducted between 1 January 2022 and 1 February 2022. The questionnaire items were introduced both in Arabic and English languages.

The questionnaire was carried out on Google Forms and distributed through different social media platforms (Facebook and WhatsApp groups). A total of 426 general populations in variable areas in Egypt filled out the questionnaire that required 3 to 5 min to complete, as shown in Figure 1.

The questionnaire consisted of 25 questions for a total possible score of 0 to 25, to determine the participants’ general awareness and preferences about the available COVID-19 vaccines in Egypt. Participants were collected through variable online providers for different governorates to avoid any bias and to achieve a broad representative sample. Inclusion criteria were available at the beginning of the survey as related to the questions that were about the demographic data of participants.

Participation in this study was voluntary, and participants received no compensation in return. Population with age below 18 years and those who rejected to participate in this survey were excluded from the study. Participation was entirely voluntary. The anonymity and confidentiality of participants were guaranteed during the data collection process as the study was conducted through a web-based anonymous survey.

### 2.2. Sample Size

The sample size was calculated using the following equation [19]:$$\;n=Z^{2}\frac{\;P\left(1-P\right)}{d^{2}}$$

(n): The required sample size;

(Z): Statistic corresponding to level of confidence;

(P): Expected prevalence;

(d): Denotes precision. Assuming an expected primary outcome (Preference of Astrazeneca Vs Sinopharm Vaccines) of 50% based on published literature [20], with a margin of error ±0.5% and a confidence level of 95%, the estimated required sample size was 384. Sample size calculation conducted using Epi-Info 7.2.3.0 software statistical package that was created by Center for Disease Control and Prevention, Atlanta, GA, USA.

### 2.3. Study Tools

This study’s tools included a questionnaire with two components. (i) Sociodemographic data involved 6 questions about gender, age, education, occupation, marital status, and place of residence. (ii) Attitude, preferences, knowledge, and practices toward COVID-19 vaccine. Nineteen questions were about knowledge, and the awareness part included eleven questions about the difference in the vaccine mechanism of action, protection against the new variant COVID-19 strains, efficacy, safety in pregnant and breastfeeding women, and the availability of vaccines for subjects less than 18 years old. There were five questions about the post-vaccine syndrome, side effects, their duration of action, and the possibility of taking analgesics with vaccination. Two questions showed the immunity produced from vaccination, and one question indicated the storage degrees of vaccines. Most of these questions were answered on a Yes/No with “I don’t know” options.

### 2.4. Descriptive and Inferential Statistics

Numbers and frequencies were used to describe nominal variables in the survey. Associations between key participant characteristics and outcomes were examined in univariable analyses using chi-square tests, fisher-exact, or Lambda test of association (as appropriate). The preference question for AstraZeneca versus Sinopharm vaccination is incorporated in a binary logistic model as a dependent variable, where age, gender, residence, educational level, and marital status are used as independent variables. Multinomial logistic regression modeling was performed for questions having more than two outcomes (Yes, No, Don’t know), where the correct answer is used as the reference category and all respondents’ characteristics are defined as model predictors, with a significance level of 0.05. Multinomial logistic regression modeling was conducted for each question’s correct answer against all respondents’ characteristics as predictors, with a significance level of 0.05. Statistical analyses were conducted using IBM SPSS v21.

## 3. Results

A total of 426 respondents were included in our study; respondents’ demographics are summarized in numbers and percentages in Table 1. Females represent about 62% of participants, the age group ranging from 25 to 35 years represents 40% of respondents, about 63.4% of respondents are living in urban communities with the majority of married marital status (61.5%), about 73% of participants have pursued a high university educational degree and about 23% of total respondents are healthcare workers.

Results of the survey questions addressing each domain are listed as follows:

### 3.1. Vaccine Preference

Vaccine preferences of AstraZeneca and Sinopharm vaccines were 50.23% and 49.76%, respectively, among all respondents as seen in the Table 2. Vaccine preference for the Sinopharm or AstraZeneca vaccines was tested using univariate association tests (Lambda association between Gender, residence, marital status and vaccine preference shows no statistically significant associations having (r) values of zero, with all participants’ characteristics, and showed no significant difference in vaccine preference according to age, gender, residence, educational level, or social status).

### 3.2. Vaccine Protection against New Variant Strains

About 50% of respondents mentioned that both AstraZeneca and Sinopharm vaccines do not offer protection against new variant COVID-19 strains, as mentioned in Table 3.

A multinomial logistic regression model was used to predict the “No” answer from respondents’ characteristics, where educational level and profession type were the only significant predictors in the model. Respondents tend to select the “Don’t know” answer as the level of education gets higher with an odds ratio of 2.7 (CI: 1.56–4.63). Healthcare workers are the lowest respondents to agree that vaccines offer protection against new COVID-19 variants (10.9%) compared to unemployed respondents (20.3%) and other professions (68.8%) with a statistically significant difference (*p* < 0.005), as represented in Table 4.

### 3.3. Effectiveness, Safety, Storage, Mechanism of Action and Side effects of Sinopharm and AstraZeneca COVID-19 Vaccines

Numbers and frequencies of respondents on questions addressing each domain are presented in Table 5. Multinominal logistic regression showed no significant difference between any of the respondents’ characteristics and having the proper answer regarding COVID-19 infection following vaccination.

However, female gender and higher education levels are significant predictors for selecting the “Don’t know” option over “No” response for immediate immunity after vaccination question with odd ratios (Adjusted OR = 2.5: CI = 1.042–5.84 and Adjusted OR = 3: CI = 1.44–6.3), respectively, where Nagelkerke r^2^ of the model = 0.13.

Residents of the rural community are less aware of proper cautious actions to take in case of prolonged side effects after vaccination (Q6) compared to urban residents with an odds ratio of (Adjusted OR = 0.27 (CI = 0.1–0.73)), where the model’s Nagelkerke coefficient r^2^ = 0.29.

Regarding the administration of analgesics, it was found that residence, education level, and profession are significant predictors in awareness regarding analgesics administration during vaccination. Higher educational level and urban residence tends to decrease the likelihood of analgesics administration before vaccination compared to after vaccination (Adjusted OR = 0.52: CI = 0.3–0.9) and (Adjusted OR = 1.9: CI = 1.04–3.38), respectively, with a model’s Nagelkerke r^2^ = 0.22. Healthcare workers have the highest rate (40%) to accept analgesia after vaccination compared to unemployed (17.9%) and other professions (11.2%), with a significant statistical difference (*p* < 0.001).

The safety of vaccine administration among children below 18 (Q8) showed statistically significant difference for gender and educational level predictors, where females are less likely to accept children below 18 vaccination compared to males (Adjusted OR= 0.48, CI: 0.25–0.95), Nagelkerke r^2^ = 0.21, while higher education tends to increase the likelihood to answer “don’t know” for children vaccination compared to rejecting children vaccination with odds ratios (Adjusted OR = 1.73: CI = 1.03–2.9), Nagelkerke r^2^ = 0.22.

The follow-up question to address the safety of vaccine administration among populations above 18 years old (Q15) showed no significant respondents’ characteristics predictor impacting the answer in the multinomial regression model with a Nagelkerke r^2^ of 0.21.

## 4. Discussion

Worldwide, 2019 coronavirus disease (COVID-19) is characterized by great progression in both its mortality rate and its incidence, yet there is no approved treatment for it [21,22]. Consequently, COVID-19 vaccines are now the life collar for all humanity to control the spread of this global epidemic. Understanding the efficacy and increasing trust in COVID-19 vaccines is the duty of the campaign concerned with vaccine education [23].

The questions included in the survey covered many significant items regarding vaccine preference, protection against new variant strains, effectiveness, safety, storage, mechanism of action, and side effects of Sinopharm and AstraZeneca COVID-19 vaccines.

Regarding the respondents’ demographics, higher participation ratios were demonstrated in females, age groups ranging from 25 to 35 years, married respondents living in urban communities, respondents with a high university educational degree, and professions other than healthcare workers. The previous online survey, carried in Ecuador, supported our findings as it recommended the vaccination campaign for people in rural areas and with a lower level of education [23].

Approximately, 50% of respondents mentioned that both AstraZeneca and Sinopharm vaccines do not offer protection against new variant COVID-19 strains. This needs to raise awareness of all populations about the effectiveness of vaccines against the Delta strain of coronavirus or any other strains. The previous study showed that AstraZeneca’s vaccine is still effective against symptomatic COVID-19 infection (33% protection after the first dose, 88% protection after the second dose) and hospitalization (to about 94% after the first dose, to about 96% after the second dose) caused by the Delta variant [24,25].

Nearly 60% of the respondents did not agree that COVID-19 vaccines cause coronavirus infection but the rest of them agreed. This means that there is a great need for a vaccine education campaign to explain the mechanism by which all vaccines act. Mostly, COVID-19 vaccines induce immune responses against COVID-19 spike protein by the production of neutralizing antibodies (NAbs), confirming that vaccines can never cause viral infection [13,26].

About 83% of the respondents agreed that most COVID-19 vaccine adverse effects may take place within the first 3 days after vaccination, mostly within the first 2 days. This represents a reasonable level of awareness among the population regarding the adverse effects of the COVID-19 vaccine compared to a previous cross-sectional study that suggested that there is insufficient background about vaccine adverse effects among the Egyptian population [27]. The majority of respondents agreed that there is no COVID-19 vaccination accessible in Egypt for children under the age of 18 and that it is available for people above the age of 18, but they had differing views on whether or not there is a COVID-19 vaccine available for pregnant and nursing mothers. Most clinical trials, concerned with the COVID-19 vaccine, did not present sufficient data on efficacy and safety for children and pregnant women. Till now, symptomatic COVID-19 poses a great danger for pregnant women to evolve into severe illness compared to non-pregnant women. Children are still apart from symptomatic COVID-19 but with the same infection rates, so they are the main source of viral transmission [28,29,30,31,32].

The participants did not agree together about whether it is correct to administer the vaccine to people who have previously contracted COVID-19. The immune memory may last for months following COVID-19 infection although the level of antibodies decreases progressively. Therefore, previously infected people may develop sustained responses rapidly compared to naïve ones. This means that sufficient response could be attained in previously infected individuals after a single dose of the vaccine [33,34,35].

About 74.6% of the respondents agreed that both Sinopharm and AstraZeneca vaccines should be stored at 2–8 °C. This represents a good knowledge of the participants about the storage temperature of the two vaccines, as this is the most suitable temperature for appropriate storage of them [36].

About half of respondents agreed that the Sinopharm vaccine is an inactivated vaccine that stimulates the human body to produce antibodies against coronavirus and that the AstraZeneca vaccine is developed using a recombinant vector technique that stimulates the human body to produce antibodies against coronavirus. However, the responses of the remaining respondents ranged from incorrect to unknown. This indicates that there is little knowledge regarding the mechanism by which vaccines are developed, highlighting the need of vaccine awareness.

Since October 2020, there have been about 212 COVID-19 candidate vaccines that are being developed worldwide. With the techniques of protein subunit vaccines, DNA-based vaccines, or RNA-based vaccines, virus-like particle development varied from inactivated vaccines to live attenuated vaccines and replicating or non-replicating viral vector vaccines [37]. Thus, it is very necessary to raise awareness of all populations about different methods of development of all COVID-19 vaccines.

Nearly all of the responses concurred that most COVID-19 patients have protection against reinfection for 5 months or so. The findings of a prior study, which showed that antibodies were still detectable in the blood of people who had recovered from SARS-COV-2 infection for roughly five to seven months after their infection, corroborated the perceptions of our respondents [38]. Another retrospective study, in the United States, approved that previous infection with COVID-19 can play an integral role in decreasing the rate of reinfection and providing protective effectiveness, depending on the immune response that may last for about 3 months [39].

The majority of the respondents agreed that vaccine effectiveness duration ranges from 6 to 12 months. However, in fact, more time is required to know how long the protection of the elicited vaccine will last and how many booster injections are required to retain fully active protection. As result, the kind of protection, either from mild or severe COVID-19 infection or mortality, can be clarified for all vaccines over time through population data [40].

The level of awareness on COVID-19 vaccines must be increased among people with lower levels of education and rural residents. This can be accomplished through media, government, and hospital-based health care team awareness efforts.

## 5. Conclusions

The findings of this study seem to add credibility to the hypothesis that the fight against the COVID-19 pandemic presents a golden opportunity to broaden participants’ knowledge of the value of vaccines. The Egyptian population in the study possessed enough understanding of coronavirus and both Sinopharm and Astrazeneca vaccines. Although the participants were pleased with the vaccine’s acceptability, there are reservations regarding it due to a lack of clinical testing and a fear of its negative effects. Moreover, it is important to provide adequate information regarding vaccines for rural areas and people with low levels of education. Future vaccination campaign methods should be designed to accommodate the fears and anxieties of participants, as well as complement public health and educational activities to increase vaccine awareness.

### Strengths and Limitations

We have successfully engaged and maintained a large cohort of dedicated participants despite the challenges of rapid implementation of a digital platform, less familiarity of our target population with electronic platforms and changing vaccine distribution timelines. On other hand, further studies may be required on larger sample sizes and on other different types of COVID-19 vaccines. Additionally, further studies may be required to examine the preferences of the higher risk group, which is the aged population over the age of 65.

## Figures and Tables

**Figure 1 ijerph-19-16803-f001:**
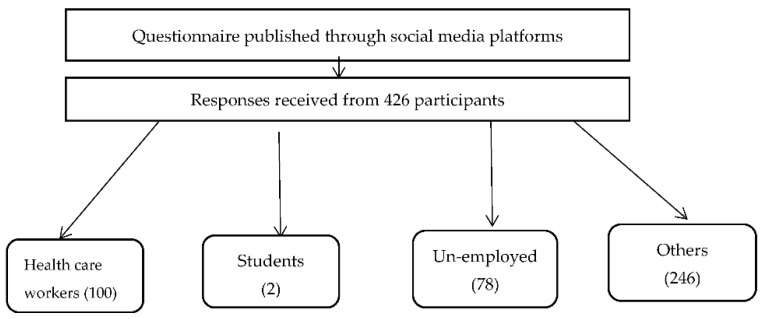
Flow chart of 426 participants in the survey.

**Table 1 ijerph-19-16803-t001:** Participants’ demographics.

Characteristics	No. (%)
Age Group	
18–25 years	58 (13.6%)
25–35 years	170 (39.9%)
35–45 years	114 (26.7%)
45–55 years	58 (13.6%)
Above 55 years	26 (6.1%)
Gender	
Male	162 (38.02%)
Female	264 (61.98%)
Residence Location	
Urban	270 (63.38%)
Rural	156 (36.62%)
Social Status	
Single	122 (28.63%)
Married	262 (61.50%)
Divorced	24 (5.63%)
Widowed	18 (4.23%)
Education Level	
Mid-level Education	52 (12.20%)
High level Education	312 (73.23%)
Post Graduate Education	62 (14.55%)
Profession	
Don’t Work	78 (18.30%)
Healthcare Worker	100 (23.47%)
Student	2 (0.47%)
Others	246 (57.74%)

**Table 2 ijerph-19-16803-t002:** Vaccine preference among respondents.

1- Which available vaccine do you prefer (Sinopharm or AstraZeneca)?	
AstraZeneca	214 (50.23%)
Sinopharm	212 (49.76%)

**Table 3 ijerph-19-16803-t003:** Respondents’ opinions regarding vaccine protection against new variant strains.

2- Do available vaccines in Egypt protect against variant strains?	
No	220 (51.64%)
Don’t Know	78 (18.30%)
Yes	128 (30.04%)

**Table 4 ijerph-19-16803-t004:** Comparison between different professions’ opinions regarding vaccine protection against new variant strains.

	**Response**	**Disagree No. (%)**	**Don’t Know No. (%)**	**Agree No. (%)**	**Total No. (%)**
**Profession**					
Don’t Work		34 (15.45%)	18 (23.07%)	26 (20.31%)	78 (18.3%)
Healthcare Worker		48 (21.81%)	38 (48.71%)	14 (10.93%)	100 (23.47%)
Others		138 (62.72%)	20 (25.64%)	88 (68.75%)	246 (57.74%)
Student		0 (0%)	2 (2.56%)	0 (0%)	2 (0.46%)

**Table 5 ijerph-19-16803-t005:** Awareness domains of COVID-19 Vaccination.

3- Can COVID-19 vaccine cause coronavirus infection?	
No	254 (59.62%)
Don’t Know	50 (11.73%)
Yes	122 (28.63%)
4- Does the COVID-19 vaccine cause immediate immunity after vaccination?	
No	290 (68.07%)
Don’t Know	40 (9.389%)
Yes	96 (22.53%)
5- Most COVID-19 vaccine side effects take place within the first 3 days after vaccination, Mostly within the first 2 days.	
Wrong	26 (6.103%)
Don’t Know	44 (10.32%)
Correct	356 (83.56%)
6- If you have side effects after vaccination for 3 days, then you should isolate yourself and do proper examinations	
Wrong	12 (2.816%)
Don’t Know	36 (8.450%)
Correct	378 (88.73%)
7- It’s acceptable to administer analgesics with vaccination	
Don’t Know	180 (42.25%)
Before Vaccination	162 (38.02%)
After Vaccination	84 (19.71%)
8- Is there any COVID-19 vaccine available in Egypt for children below 18?	
No	316 (74.17%)
Don’t Know	70 (16.43%)
Yes	40 (9.389%)
9- Do people previously infected by COVID-19 will get vaccinated?	
No	176 (41.31%)
Don’t Know	48 (11.26%)
Yes	202 (47.41%)
10- Both Sinopharm or Oxford vaccine is stored at 2–8 °C	
Wrong	30 (7.042%)
Don’t Know	78 (18.30%)
Correct	318 (74.64%)
11- Sinopharm Vaccine has effectiveness more than 86% and is given as 2 doses separated by 21 days.	
Wrong	30 (7.042%)
Don’t Know	74 (17.37%)
Correct	322 (75.58%)
12- Oxford Vaccine has an effectiveness of more than 90% and is given in 2 doses separated by 3 months.	
Wrong	32 (7.511%)
Don’t Know	94 (22.06%)
Correct	300 (70.42%)
13- The Sinopharm vaccine is an inactivated vaccine that stimulates the human body to produce antibodies against coronavirus.	
Wrong	44 (10.32%)
Don’t Know	156 (36.61%)
Correct	226 (53.05%)
14- AstraZeneca vaccine is developed through recombinant vector technique that simulates the human body to produce antibodies against coronavirus.	
Wrong	42 (9.859%)
Don’t Know	166 (38.96%)
Correct	218 (51.17%)
15- The vaccine is available for populations above 18 years old.	
Wrong	16 (3.755%)
Don’t Know	32 (7.511%)
Correct	378 (88.73%)
16- Most patients infected by the COVID-19 virus, have immunity against reinfection for 5 months.	
Wrong	38 (8.920%)
Don’t Know	48 (11.26%)
Correct	340 (79.81%)
17- Is it possible to get reinfected after vaccination?	
Wrong	28 (6.572%)
Don’t Know	46 (10.79%)
Correct	352 (82.62%)
18- Vaccine effectiveness duration ranges from 6 to 12 months.	
Wrong	18 (4.225%)
Don’t Know	80 (18.77%)
Correct	328 (76.99%)
19- Is there any COVID-19 vaccine available for pregnant and breastfeeding women?	
No	168 (39.43%)
Don’t Know	102 (23.94%)
Yes	156 (6.61%)

## Data Availability

Available upon request from the corresponding author.

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
