# Peer review of "Knowledge, Attitude, and Acceptance of Sinopharm and AstraZeneca’s COVID-19 Vaccines among Egyptian Population: A Cross-Sectional Study"

_ijerph, 2022, doi:10.3390/ijerph192416803_

Round 1

Reviewer 1 Report

The manuscript "Knowledge, Attitude, and Acceptance of Sinopharm and AstraZeneca's COVID-19 Vaccines among Egyptian Population: A Cross-Sectional Study" by Boshra et al is an important study

The methods and the result are well presented. But as the Covid has significantly affected  the elderly population and they are unlikely to use the the social media platforms

I recommend to publish the paper after authors include atleast the data from the literature of the aged population 

Update:

The authors have investigated the Egyptian people's awareness related to COVID-19 vaccines. I think the research is relevant as it would help to evaluate different factors that would reflect general public confidence on the vaccines. The data presented is a significant addition to the known literature.    The conclusion is consistent with the results and discussion and the references are appropriate I had sent the comment that authors didn't consider the elderly population who are unlikely to respond to electronic platforms; and it is crucial especially as we know Covid-19 impacted mostly the aged population Also, it would be good for the authors to have summarized the tables in a graphical abstract format

Author Response

Thank you for your insightful comment to clarify this point. Because the data was gathered through an online questionnaire during a certain time period, it was a cross-sectional web-based study that revealed Egyptian population knowledge, attitudes, and acceptance of Sinopharm and AstraZeneca's COVID-19 vaccines. As a result, in the resubmitted version, the analysis of the preferences of the elderly population is added in the limitation section following the conclusion, and it will be taken into account in future work with other COVID-19 vaccines types.

Reviewer 2 Report

I enjoyed reading the article " Knowledge, Attitude, and Acceptance of Sinopharm and AstraZeneca's COVID-19 Vaccines among Egyptian Population: A Cross-Sectional Study”.

A clear manuscript describes in a simple and clear way how important and still relevant is the need to educate the public about immunization. The summary is consistent with the content of the work. The reviewed work presents the current state of knowledge (modern knowledge is enriched with further data helpful in the fight against the ongoing pandemic) in a logical, understandable, comprehensive, carefully and correctly citing literature - 40 items. It does not contain factual errors, the names are in accordance with generally accepted principles. In my opinion, the text in English does not contain language errors. 

Overall assessment of work very good - this work is valuable and leads to positive conclusions presented by the authors.

My comments:

In figure 1. (green frame) add the number of health care workers participating in the study.

Author Response

Thank you for drawing our attention, the number of health care workers participating in the study has been added in figure 1 in the resubmitted version.

Reviewer 3 Report

The paper concerns an investigation of Egyptian people knowledge and preferences concerning COVID-19 vaccines. The Authors made a web-based survey and the questions they asked concerned general knowledge about COVID-19 vaccines as well as vaccines developed by AstraZeneca and Sinopharm which were available in Egypt.

Some of the questions included in the survey are rather standard and can be found in many other similar investigations, but some other are not. For example, there are questions concerning the way in which the vaccines were developed, in what temperatures they should be stored and questions concerning the nature of immunity against SARS-CoV-2. It is interesting that a grate fraction of the survey participants correctly answered these questions (it could be expected that most people are not very interested in such  problems). So, it seems that Egyptian people have quite good knowledge about the vaccination process - it is a positive result of the survey.

However, there are also some week points of the paper. The conclusions are very short and rather negatively surprising in light of the (potentially) interesting result of the survey. In one of the two sentences of conclusions the Authors say that there is a need for educational campaigns for "rural areas and people with low education levels". This is what should be expected and similar conclusions can be drawn for many other places in the world. The Authors analyzed the obtained results in Discussion section and, in my opinion, they could summarize their research by more interesting conclusions.

Moreover, the Authors wrote that they obtained a broad representative group of participants of the survey, but 23.47% of them are healtcare workers. It is difficult to agree with the Authors that the sample is not biased.

In addition, something is wrong with figure 1 - there are two versions of the same scheme.

The symbols in the equation in page 4 are not explained.

The English language and style should be carefully checked and corrected in the whole paper.

To summarize, in my opinion the paper could be considered for a possible publication buy the Authors should take into accont the above mentioned points.

Author Response

Comment
In addition, something is wrong with figure 1 - there are two versions of the same scheme.

Authors Reply:

Thank you for drawing our attention for this comment, one version of the figure 1 has been adjusted in the resubmitted manuscript.

Comment
The symbols in the equation in page 4 are not explained.

Authors Reply:

Thank you for drawing our attention for this comment to clarify these symbols, as they are explained as following in the resubmitted manuscript.

Required sample size (n) is calculated using (Z) statistic corresponding to level of confidence, while P represents expected prevalence and d denotes precision.

Comment

The English language and style should be carefully checked and corrected in the whole paper.

Authors Reply:
 Thank you for drawing our attention for this comment, the English language and style is edited in different parts in the whole resubmitted version.

To summarize, in my opinion the paper could be considered for a possible publication buy the Authors should take into account the above mentioned points.

Round 2

Reviewer 3 Report

The Authors only partially have taken into account my previous comments  (they corrected Figure 1 and axplained the symbols used in the equation in subsection 2.2). Hence, in my opinion the paper still needs some improvements (according to the comments which they haven't addressed).

Author Response

 1- Comment

However, there are also some week points of the paper. The conclusions are very short and rather negatively surprising in light of the (potentially) interesting result of the survey. In one of the two sentences of conclusions the Authors say that there is a need for educational campaigns for "rural areas and people with low education levels". This is what should be expected and similar conclusions can be drawn for many other places in the world. The Authors analyzed the obtained results in Discussion section and, in my opinion; they could summarize their research by more interesting conclusions.

Response to reviewer

Thank you for drawing our attention to clarify this point, the conclusion is adjusted as your suggestion in resubmitted version

2- Comment
Moreover, the Authors wrote that they obtained a broad representative group of participants of the survey, but 23.47% of them are healthcare workers. It is difficult to agree with the Authors that the sample is not biased.

Authors Reply:

Thank you for your insightful comment to clarify this point. The data of the survey was gathered randomly through an online questionnaire. The number of healthcare workers did not exceed the quarter of the participants, about 18.3% of the participants did not work, and about 57.74% of them had different other jobs, which reflects no bias in our studied population. The 23.47%  of the healthcare workers is considered to be logic as a participant in the survey based on their presence in this time on social media to aware population about the pandemic.